# Preferential Tissue Sites of Different Cancer-Risk Groups of Human Papillomaviruses

**DOI:** 10.3390/ijms241713151

**Published:** 2023-08-24

**Authors:** Mitsuaki Okodo, Kaori Okayama, Toshiyuki Sasagawa, Koji Teruya, Rei Settsu, Shuichi Mizuno, Yasuyoshi Ishii, Mizue Oda

**Affiliations:** 1Department of Medical Technology, Faculty of Health Sciences, Kyorin University, 5-4-1 Shimorenjaku, Mitaka-shi 181-8621, Tokyo, Japan; rei-settsu@ks.kyorin-u.ac.jp (R.S.); mizuno1911h@std.kyorin-u.ac.jp (S.M.); 2Department of Health Science, Gunma Paz University Graduate School of Health Sciences, 1-7-1 Tonyamachi, Takasaki-shi 370-0006, Gunma, Japan; okayaman0811@std.kyorin-u.ac.jp; 3Department of Obstetrics and Gynecology, Kanazawa Medical University, 1-1 Uchinadadaigaku, Kahoku-gun, Kanazawa 920-0293, Ishikawa, Japan; tsasa@kanazawa-med.ac.jp; 4Department of Health and Welfare, Faculty of Health Sciences, Kyorin University, 5-4-1 Shimorenjaku, Mitaka-shi 181-8621, Tokyo, Japan; teruya@ks.kyorin-u.ac.jp; 5Department of Clinical Laboratory, Genki Plaza Medical Center for Health Care, 1-7-1 Jinbocho, Chiyoda-ku 101-0051, Tokyo, Japan; y-ishii@genkiplaza.or.jp; 6Department of Gynecology, Genki Plaza Medical Center for Health Care, 1-105 Jinbocho, Chiyoda-ku 101-0051, Tokyo, Japan; m-oda@genkiplaza.or.jp

**Keywords:** human papillomavirus, tropism, contamination, cervical sample, vaginal sample

## Abstract

The oncogenic potential of human papillomavirus (HPV) may be used to determine the tissue tropism of each HPV type. Cervical cancer develops in the squamo-columar junction of the cervices, and most lesions are induced by high-risk (HR) HPV types. This suggests that HR types preferentially infect the cervix, whereas the preferential infection site for low-risk (LR) types is not well defined. The determination of HPV tropism when using cytology samples can be uncertain since it is difficult to avoid contamination of cell samples between the cervix and the vagina. Herein, cell samples were carefully collected by independently scraping the cervix and vagina, after which the HPV types were determined. HPV tissue tropism was determined by considering what HPV types were positive at only one of the sites (the cervix or the vagina) as the viruses that preferentially infected that site. This method revealed that all LR types were only identified in vaginal samples, whereas 87% of HR types were identified in cervical sites. Thus, LR types may preferentially infect the vagina, whereas HR types infect the cervix. These findings suggest that preferential tissue tropism of certain HPV types is a probable factor for malignant progression.

## 1. Introduction

Approximately 50 human papillomavirus (HPV) types have so far been identified in the mucosal epithelium of the uterine cervix, vagina, and vulva [1]. Of these, the World Health Organization suggests that 12 HPVs (HPV16, HPV18, HPV31, HPV33, HPV35, HPV39, HPV45, HPV51, HPV52, HPV56, HPV58, and HPV59) are defined as high-risk (HR) types, and HPV68 is classified as a possible HR type (pHR) by epidemiological evidence up to 2009 [2]. Pathological evidence has demonstrated cervical high-grade squamous intraepithelial lesions (HSIL) and cancer development in the squamocolumnar junction (SCJ) of the uterine cervix [3,4,5]. However, the risk categories are not certain for such HPV types as HPV26, HPV30, HPV34, HPV53, HPV66, HPV67, HPV69, HPV70, HPV73, HPV82, and HPV85 that belong to the α-5, α-6, α-7, α-9, and α-11 phylogenetic classification, and these types are considered pHR types since they were identified in cervical cancer tissue [6,7,8]. HPV types that do not belong to these groups are likely to be low-risk (LR) types since these types are rarely identified in cancer, and molecular evidence for E6 and E7 proteins of these HPV types suggests a low carcinogenic potential in vivo study [9,10]. Differences in carcinogenicity are caused by diverse promoter activity, gene expression patterns, and functional distinctions between the E6 and E7 proteins [9,10].

Another mechanism for these differences in carcinogenicity may be tissue tropism [11,12]. Most cervical HSIL and cancers develop in the SCJ, or transformation zone, of the cervix, and many lesions are induced by HR-HPV types [3,4,5]. Several studies have demonstrated that HR-HPV types preferentially infect the columnar/metaplastic cells of the SCJ of the cervix as opposed to the mature squamous epithelium of the vagina [11,12,13,14,15], whereas pHR and LR types prefer the vaginal epithelium [14,16,17,18]. Thus, it is logical that HR types are strongly involved in the development of cervical lesions because of their location in the SCJ, where specific cells with stem cell-like potential exist. However, conflicting findings suggest that pHR types with a preference for vaginal epithelium can also be detected in cervical carcinoma [6,7,8,18], whereas HR types with a preference for cervical epithelium are involved in the development of high-grade vaginal intraepithelial lesions (VaIN) [19,20]. In addition, because hypothetical tropism does not imply the absolute infection of a specific site [11,12], the presence of an LR-type lacking a preference for the cervix in a cervical sample cannot be dismissed as contamination from the vagina, which is somewhat confusing considering the possibility of preclinical cervical infection or intraepithelial lesions caused by LR types [20]. The cause of these discrepancies and confusion may be attributed to previous analyses that were inaccurately performed as data were estimated from relative differences in HPV prevalence using samples of uncertain provenance with selection bias reflecting the cervical or vaginal epithelium, e.g., scraped cervical cells from hysterectomized/nonhysterectomized patients [15,16].

Therefore, this study sought to define the HPV types that preferentially infect cervical or vaginal tissues by eliminating cross-contamination between sites from HPV genotyping results of cytology samples from the cervix, upper vaginal, and lower vaginal tissues. Our results support the hypothesis that LR types preferentially infect vaginal tissue and that HR types preferentially infect cervical tissue, while pHR types do not display a tissue preference.

## 2. Results

The prevalence and exact 95% CIs of HPV types in the cervical and vaginal (upper or lower) samples from the 264 HPV-positive cases are shown in Table 1. The common HPV types in the cervix were HPV52 (25.0%) and HPV51 (13.6%). The most common HPV types in the upper vagina were HPV52 (25.0%) and HPV58 (12.9%), and in the lower vagina were HPV52 (25.0%) and HPV74 (8.7%). The prevalence of HPV tends to decrease with distance from the cervix. Chi-squared tests of independence between each sample and HPV type, HPV risk group, and α phylogenetic group did not identify any significant associations between these groups.

All HPV types detected in the 86 patients with inconsistent HPV type results between cervical and vaginal samples are shown in Table 2. The mean age of these patients was 40 years old (range, 24–65 years old). To eliminate as much contamination between sites as possible, HPV types detected in only one of the samples were identified as the HPV type that preferentially infected that site. The prevalence of HPV types with cervical or vaginal preference in the HPV risk groups and α-phylogenetic groups is shown in Figure 1 A strong and significant association was found between the HPV risk group and cervical and vaginal preference (χ^2^(2) = 52.612, *p* < 0.01). HR types were significantly more common in the cervix than in the vagina, with 87% preferentially infecting the cervix (*p* < 0.01). HR types were 6.7-fold more common in the cervix than in the vagina. All LR types preferentially infected the vagina (*p* < 0.01). The prevalence of pHR types was comparable between the cervix and the vagina. A strong and significant association with the α phylogenetic group was also found (χ^2^(4) = 47.375, *p* < 0.01). The α-5 and α-9 phylogenetic groups preferentially infected the cervix over the vagina, 91.7% (*p* < 0.05) and 86.8% (*p* < 0.01), respectively. The α-5 and α-9 phylogenetic groups were 11.0- and 6.6-fold more frequent in the cervix than in the vagina, respectively. The α-11/α-1/α-8/α-10/α-13/α-3/α-15 phylogenetic groups all preferentially infected the vagina (*p* < 0.01).

Because a cervical abrasion sample used for cervical cancer screening is often assumed to contain only cervical cells, assuming that LR types preferentially infect the vagina and HR types preferentially infect the cervix, the colocalization rates (including possible contamination) of different HPV risk groups in 178 cervical and vaginal samples with HPV-type concordance in each sample are shown in Table 3. The colocalization rates of the LR-type infected cells from the vagina to the cervix and the HR-type infected cells from the cervix to the vagina were 34.8% and 73.6%, respectively.

## 3. Discussion

Herein, we used a highly sensitive HPV polymerase chain reaction (PCR) method to study HPV tissue tropism in cell samples from the cervix and vagina. Our results suggest that the LR and HR types have a significant preference for vaginal and cervical epithelium, respectively, while the pHR type displays no tissue preference. This study strongly supports the hypothesis that mucosal HPVs have tropism for specific sites within the lower genital tract.

Prior studies of HPV tropism compared women with hysterectomies to those without [15,16] and women younger than 50 to those 50 and older [14]. These comparisons were made on the basis that with the former population, the cervical region is abraded for the collection of columnar and metaplastic cells of the cervical SCJ, whereas with the latter populations, only cells from the mature squamous epithelium similar to that of the vagina were collected. However, age-related sampling and hysterectomy-related potential biases may have affected these study results. To eliminate such biases, Castle et al. [17] collected paired vaginal and cervical specimens from the same patients and examined the relationship between vaginal and cervical HPVs at the type-specific level. They revealed that noncarcinogenic HPV types of the α-3/α-15 and α-1/α-8/α-10 phylogenetic groups, which corresponded to LR types, may have tropism for the vaginal epithelium, whereas HR types of the α-9/α-11 groups appeared to have similar affinity for both the vaginal and cervical epithelium. Although their study was similar to our analysis in terms of sampling methods, our results showed a clear preference for HR HPVs, or α5/α9 groups, for cervical infection and that of LR HPVs, or α-11/α-1/α-8/α-10/α-13/α-3/α-15 groups, for vaginal infection. The reported susceptibility of both tissues to the α-9/α-11 groups [17] is also explained by the cervix and vagina being preferentially infected by the α-9 and α-11 groups, respectively.

We consider that our results differ in part from those of previous studies because we eliminated contamination by vaginal cells in the cervical samples and by cervical cells in the vaginal samples to the extent possible. Thus, the HPV detection results at the three sites showed that the identified HPV types were almost identical. The only way to reveal HPV preference in this situation is to identify HPV types that are uncommon in each sample. We believe that this method of tropism analysis is why we can clarify the tissue susceptibility of each risk group.

Tropism is accepted as a preference for a particular anatomical site and is not considered to be directly related to the development of cancer at that site [11,12]. This supposition is supported given that LR types prefer the vagina but rarely induce the development of vaginal lesions. E6 and E7 proteins of the LR-type are not reportedly involved in the development of HSIL because they do not drive host cell proliferation or transform these cells into cancer cells [21]. We previously reported that no LR types caused a single-type infection in patients with HSIL and cancer of the cervix [22]. In addition to functional differences in E6 and E7 protein activity between the LR and HR types, the LR-type does not preferentially infect cervical tissue, which might explain why the LR HPV types are less likely to cause persistent infection and induce the development of HSIL.

Because the genotypes of pHR HPVs have a low prevalence in patients with cervical cancer, consensus has not emerged to classify these HPVs as HR types [23]. The low prevalence of pHR types in the cervix is probably explained by the lack of susceptibility of the cervix to genotypes of pHR HPVs, as all α-11 genotypes (HPV34 and HPV73) displayed a preference for the vaginal epithelium. pHR types have been demonstrated to preferentially infect the vaginal epithelium [18], which is inconsistent with their involvement in the development of high-grade cervical lesions. This may be explained here, as we identified that α-5 (HPV26 and HPV82) and α-9 (HPV67) in the pHR group exhibited tropism for the cervix.

A cervical abrasion sample used for cervical cancer screening is often assumed to contain only cervical cells. However, assuming that LR types preferentially infect the vagina, our colocalization analysis revealed that approximately 30% of cervical specimens contained a mixture of LR type-infected cells of vaginal origin. In cervical cancer screening, a preclinical cervical infection is often determined when the Hybrid Capture II (HC) assay is positive and the biopsy is negative [24]. Although the specificity of HC varies depending on whether the endpoint is cervical intraepithelial neoplasm (CIN) 2+ or not, some of the reported 7–25% false positives [25] may require consideration of possible contamination of LR-type infected cells from the vagina to the cervix, given that HC cross-reacts with LR-type and becomes positive [26]. Conversely, the vagina is considered the reservoir for any mucosal HPV types [20], and this may be because LR types preferentially infect the vagina while vaginal accumulation of cervix-derived HR and pHR type-infected cells occurs during most phases of the menstrual cycle [27]. Colocalization analysis also revealed HR types in approximately 70% of the vaginal samples, presumably caused by cervical-to-vaginal contamination. Under vaginal infection by various HPV types, the HR type is mainly known to cause high-grade VaIN and vaginal cancer [19]. Previously, we used manual microdissection of lesions from formalin-fixed, paraffin-embedded tissue specimens to demonstrate that the HR or pHR type is responsible for high-grade VaIN and squamous cell carcinoma [20]. However, the prevalence of vaginal lesions is significantly lower than that of cervical lesions, with an annual incidence of <1 per 100,000 women [28]. This confirms our results that HR types are less likely to preferentially infect the vagina. By contrast, women with a history of CIN2+ have been reported to have higher frequencies of VaIN and vaginal cancer because of persistent infection by HR types [29,30]. Thus, if HR types contribute to the development of cervical lesions, they would also infect the vaginal epithelium. Nevertheless, we recently found that HPV types from VaIN and CIN lesions in the same patient differed in 92.3% (24/26) of cases, suggesting that VaIN and CIN develop independently [20]. Therefore, the higher risk of VaIN in patients with CIN is not because of the HPV strains that contribute to the development of CIN, as these are more likely to be transmitted because they prefer the vagina. Instead, the immune response of the host [31,32], who is prone to acquiring persistent infection, is similarly impaired in the vagina. Consequently, we believe that HPV tissue tropism is not a significant factor in the development of vaginal lesions but due to the success of any of the various HR or pHR types in the vaginal reservoir in establishing a persistent infection.

This study had some limitations. First, our statistics and inferences were restricted by the limited number of samples included in this study. Furthermore, this study only speculated on the tropism of HPV for the cervix and vagina based on HPV genotyping using cell samples obtained from a small population of women. In addition, our selection criteria for this study may have biased our findings. Unfortunately, this study did not have accurate HPV type data for the biopsied tissues, and we could not assess whether each HPV risk group truly exhibited tropism for each tissue. Nonetheless, given that similar results were obtained in studies using paired scraped cervical and vaginal samples, further research is warranted because additional data could further clarify HPV tropism and the development of cervical and vaginal lesions.

## 4. Materials and Methods

### 4.1. Patients and Specimen Collection

A total of 324 randomly selected patients who had been referred for colposcopy because of abnormalities identified during cervical cytologic screening at Tokyo Genki Plaza Health Medical Center from 2020 to 2021 were enrolled in the study. Three types of SurePath™ (Becton Dickinson and Company, Franklin Lakes, NJ, USA) liquid-based cytology (LBC) were used to collect samples from the cervix and the upper and lower vaginal tissues of the patients.

Immediately before colposcopic assessment, a vaginal speculum was used to expose the cervix of each patient, and then the endocervix and portio vaginalis were scraped using an IM sampler instrument (Muto Pure Chemical, Tokyo, Japan) consisting of a spatula and brush and eluted into a collection vial, which was used as the cervical sample. Then, while holding the vaginal speculum open, the bilateral portions from the vaginal vestibule to the upper half of the vaginal canal were scraped with a spatula and collected as the upper vaginal sample, and cells from the lower half of the vaginal canal were collected in the same manner as the lower vaginal sample. Samples were collected after obtaining written informed consent from the patients. The Ethics Committee on Human Research of Kyorin University approved the study protocol, which was implemented in accordance with the approved guidelines. Patients who were negative for HPV in three different samples were excluded, and consequently, 264 patients who were positive for any HPV type in any of the samples were evaluated (Figure 2). The patients were nonpregnant women with a mean age of 40 (range, 22–71) years. Several samples used in the present study had previously been reported in a study of the effects of the menstrual cycle on the accumulation of HPV-infected cells exfoliated from the cervix that drift into the vagina [27]. 

### 4.2. HPV Genotyping Using LBC Samples

DNA was isolated from the cell pellet of each LBC sample using the hot sodium hydroxide method [33]. Cell pellets were lysed with 50 μL of alkaline lysis solution (25 mM NaOH and 0.2 mM ethylenediaminetetraacetic acid; pH 12.0) for 30 min at 95 °C. Lysed cells were then neutralized with 0.04 M Tris-HCl (pH 5.0), centrifuged at 13,200 rpm for 1 min, and directly used as the DNA template for PCR amplification. All HPV-positive samples tested positive for human β-globin DNA, demonstrating that DNA of amplifiable quality was extracted from the specimens. HPV genotyping was performed using a highly sensitive HPV PCR method known as uniplex E6/E7 PCR [34]. This method can identify the E6 or E7 genes of 39 mucosal HPV types, including 12 HR types (HPV16, HPV18, HPV31, HPV33, HPV35, HPV39, HPV45, HPV51, HPV52, HPV56, HPV58, and HPV59), 12 pHR types (HPV26, HPV30, HPV34, HPV53, HPV66, HPV67, HPV68, HPV69, HPV70, HPV73, HPV82, and HPV85), and 15 LR types (HPV6, HPV11, HPV40, HPV42, HPV44, HPV54, HPV55, HPV61, HPV62, HPV71, HPV74, HPV81, HPV84, HPV89, and HPV90), from as few as 100 viral copies, with no cross-reactivity across all HPV genotypes.

### 4.3. Data Analysis

Prevalence estimates and exact 95% CIs were calculated for each individual HPV type, HPV risk group, and α phylogenetic group detected in cervical, upper vaginal, and lower vaginal samples from 264 patients, and HPV types that preferred the cervix or vagina were then evaluated from the individual HPV types detected in the 86 patients with inconsistent HPV types in the vaginal (upper and lower) samples versus the cervical samples. HPV types detected in only one of the samples were identified as the HPV types that preferentially infected that site. For example, if HPV56 and HPV62 were positive in the cervical sample and only HPV62 was positive in the vaginal upper or lower samples, HPV62 was considered to be a virus that had migrated from one site and contaminated the other; HPV56 was identified as the only virus that preferentially infected the cervix. Differences in the prevalence of each HPV type were compared by using a z test for the difference between two proportions with Bonferroni adjustment for multiple comparisons using Statistical Package for the Social Sciences version 25.0 (SPSS Inc., Chicago, IL, USA). Furthermore, we investigated the colocalization rate, including possible contamination of HPV types with vaginal tropism in the cervical samples and with cervical tropism in the vaginal samples based on the results of tropism analysis.

## 5. Conclusions

This study evaluated HPV tissue tropism and eliminated cross-contamination of HPV genotyping results using cervical and vaginal cytology samples. The results emphasize that LR types preferentially infect vaginal tissue, while HR types preferentially infect cervical tissue. Thus, preferential tissue tropism of certain HPV types is a probable factor for malignant progression.

## Figures and Tables

**Figure 1 ijms-24-13151-f001:**
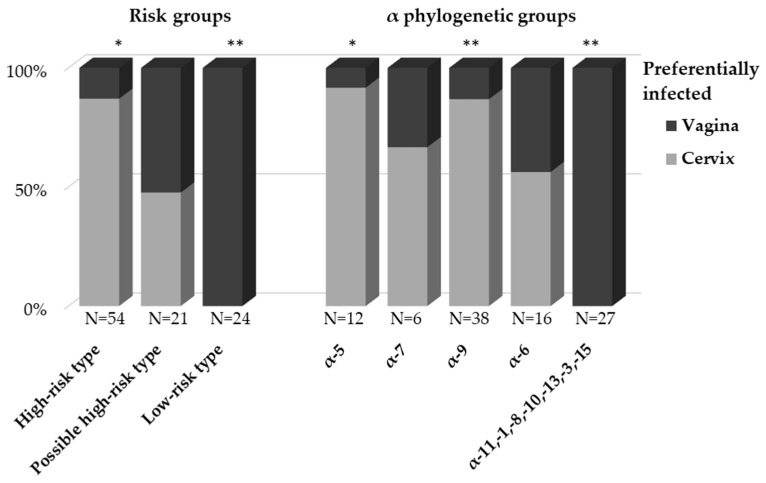
Prevalence of HPV that preferentially infected the cervix or vagina in HPV risk groups and α-phylogenetic groups. ** *p* < 0.01, * *p* < 0.05.

**Figure 2 ijms-24-13151-f002:**
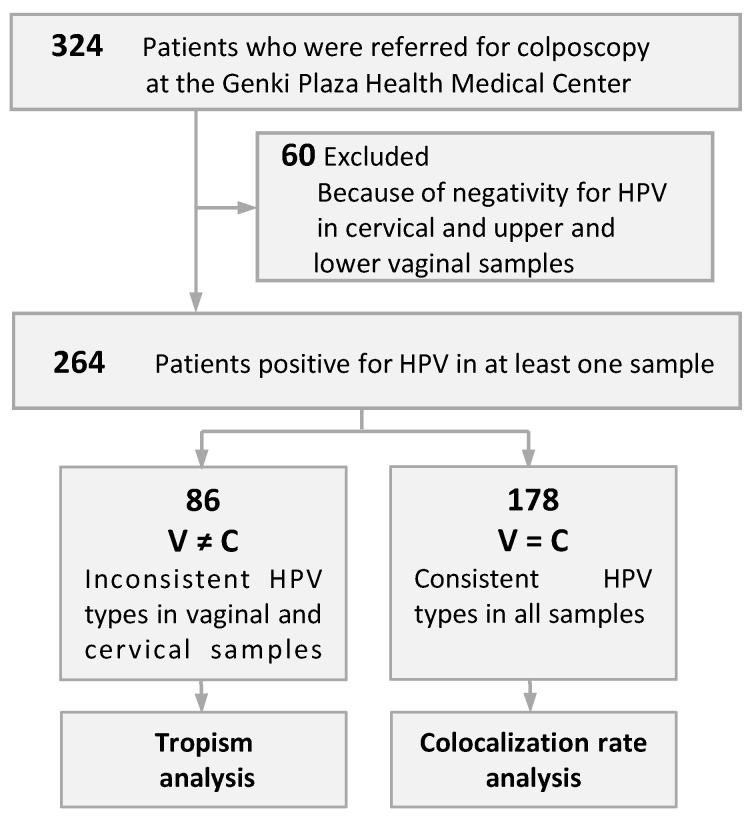
Stratification of patients evaluated for cervical and vaginal tropism and contamination of cervical samples by human papillomavirus (HPV) with vaginal tropism.

**Table 1 ijms-24-13151-t001:** Prevalence of all HPV types detected in cervical samples and upper or lower vaginal samples from 264 patients.

	Prevalence (95% CI) %		
	Cervical Samples	Vaginal Samples	
		Upper	Lower
HPV types			
HPV16	12.9 (8.8–16.9)	10.6 (6.9–14.3)	4.5 (2.0–7.1)
HPV18	1.9 (0.2–3.5)	1.1 (0.0–2.4)	0
HPV31	9.5 (5.9–13)	7.6 (4.4–10.8)	2.7 (0.7–4.6)
HPV33	2.7 (0.7–4.6)	2.3 (0.5–4.1)	1.1 (0.0–2.4)
HPV35	0.4 (0.1–1.1)	0.4 (0.0–1.1)	0.4 (0.0–1.1)
HPV39	3.8 (1.5–6.1)	3.4 (1.2–5.6)	1.9 (0.2–3.5)
HPV45	1.1 (0.0–2.4)	1.1 (0.0–2.4)	0.8 (0.0–1.8)
HPV51	13.6 (9.5–17.8)	10.2 (6.6–13.9)	4.5 (2.0–7.1)
HPV52	25.0 (19.8–30.2)	19.3 (14.6–24.1)	11.4 (7.5–15.2)
HPV56	9.5 (5.9–13.0)	8.0 (4.7–11.2)	5.3 (2.6–8.0)
HPV58	12.9 (8.8–16.9)	12.1 (8.2–16.1)	5.3 (2.6–8.0)
HPV59	4.2 (1.8–6.6)	4.5 (2.0–7.1)	3.4 (1.2–5.6)
HPV26	0.4 (0.0–1.1)	0	0
HPV30	0.8 (0.0–1.8)	0.8 (0.0–1.8)	0.4 (0.0–1.1)
HPV34	1.9 (0.2–3.5)	2.3 (0.5–4.1)	1.1 (0.0–2.4)
HPV53	8.0 (4.7–11.2)	8.3 (5.0–11.7)	4.2 (1.8–6.6)
HPV66	8.3 (5.0–11.7)	8.3 (5.0–11.7)	5.7 (2.9–8.5)
HPV67	3.8 (1.5–6.1)	2.7 (0.7–4.6)	1.5 (0.0–3.0)
HPV69	1.1 (0.0–2.4)	1.1 (0.0–2.4)	0.8 (0.0–1.8)
HPV68	3.0 (1.0–5.1)	3.0 (1.0–5.1)	1.5 (0.0–3.0)
HPV70	1.5 (0.0–3.0)	0	0.8 (0.0–1.8)
HPV73	0	0.4 (0.0–1.1)	0
HPV82	7.6 (4.4–10.8)	6.8 (3.8–9.9)	0
HPV6	1.5 (0.0–3.0)	1.5 (0.0–3.0)	1.5 (0.0–3.0)
HPV11	0.4 (0.0–1.1)	0.4 (0.0–1.1)	0
HPV40	2.7 (0.7–4.6)	3.0 (1.0–5.1)	0
HPV42	6.4 (3.5–9.4)	7.2 (4.1–10.3)	4.5 (2.0–7.1)
HPV44	0.4 (0.0–1.1)	0.4 (0.0–1.1)	0.4 (0.0–1.1)
HPV54	3.4 (1.2–5.6)	3.8 (1.5–6.1)	1.9 (0.2–3.5)
HPV55	1.9 (0.3–3.5)	2.3 (0.5–4.1)	1.9 (0.2–3.5)
HPV61	3.4 (1.2–5.6)	3.4 (1.2–5.6)	1.1 (0.0–2.4)
HPV62	4.9 (2.3–7.5)	5.7 (2.9–8.5)	2.7 (0.7–4.6)
HPV71	2.7 (0.7–4.6)	2.7 (0.7–4.6)	0.4 (0.0–1.1)
HPV74	8.7 (5.3–12.1)	9.8 (6.3–13.4)	5.7 (2.9–8.5)
HPV81	3.4 (1.2–5.6)	4.2 (1.8–6.6)	1.9 (0.2–3.5)
HPV84	0.8 (0.0–1.8)	1.5 (0.0–3.0)	1.1 (0.0–2.4)
HPV89	0.8 (0.0–1.8)	0.8 (0.0–1.8)	0.4 (0.0–1.1)
HPV90	4.5 (2.0–7.1)	5.3 (2.6–8.0)	3.4 (1.2–5.6)
Risk groups			
High-risk type	95.1 (92.5–97.7)	80.7 (75.9–85.4)	41.3 (35.3–47.2)
Possible high-risk type	34.5 (29.1–40.6)	35.2 (29.5–41.0)	17.4 (12.8–22.0)
Low-risk type	39.0 (33.1–44.9)	50.4 (44.3–56.4)	26.9 (21.5–32.2)
α phylogenetic groups			
α-5	21.6 (16.6–26.6)	18.2 (13.5–22.8)	6.8 (3.8–9.9)
α-7	15.2 (10.8–19.5)	14.8 (10.5–19.1)	8.3 (5.0–11.7)
α-9	65.2 (59.4–70.9)	54.9 (48.9–60.9)	26.9 (21.5–32.2)
α-6	25.8 (20.8–31.4)	25.4 (20.1–30.6)	15.5 (11.2–19.9)
α-11/α-1/α-8/α-10/ α-13/α-3/α-15	40.9 (35.0–46.8)	53.0 (47.0–59.1)	28.0 (22.6–33.4)

HPV, human papillomavirus.

**Table 2 ijms-24-13151-t002:** HPV types that preferentially infected the cervix or vagina.

Cases	Age	HPV Types		All HPV Types Detected		
		Preferentially Infected the Cervix	Preferentially Infected the Vagina	Cervical Samples	Vaginal Samples	
					Upper	Lower
1	51	56 (HR, α-6)	−	56, 62	62	Negative
2	37	−	81 (LR, α-3)	Negative	81	81
3	31	31 (HR, α-9)	−	31, 81, 82	81, 82	Negative
4	34	−	16 (HR, α-9)	39, 56, 71	16, 39, 56, 71	39, 56
5	24	−	90 (LR, α-15)	59	59, 90	59, 90
6	41	16 (HR, α-9)	−	16	Negative	Negative
7	50	−	84 (LR, α-3)	Negative	84	84
8	31	52 (HR, α-9)	−	16, 45, 52, 40	16, 45, 40	16, 45, 40
9	40	−	56 (HR, α-6)	Negative	56	56
10	50	−	66 (pHR, α-6)	51, 56, 58	51, 56, 58	51, 56, 66
11	35	52 (HR, α-9)	−	52, 53, 74	53, 74	53, 74
12	26	−	59 (HR, α-7)	56, 66, 74, 90	56, 59, 66, 74, 90	56, 59, 66, 74, 90
13	40	56 (HR, α-6)	−	56	Negative	Negative
14	50	51 (HR, α-5)	−	51, 58	58	58
15	45	−	34 (pHR, α-11)	Negative	Negative	34
16	26	58 (HR, α-9)	−	52, 58	52	52
17	58	−	52 (HR, α-9)	34	52, 34	34
18	45	−	34 (pHR, α-11)	31,74	31, 34, 74	Negative
19	26	−	54 (LR, α-13), 66 (pHR, α-5), 74 (LR, α-10), 90 (LR, α-15)	56	56, 54, 66	56, 54, 66, 74, 90
20	42	39 (HR, α-7)	−	39, 58	58	58
21	43	51 (HR, α-5)	−	51	Negative	Negative
22	31	52 (HR, α-9)	−	52	Negative	Negative
23	47	16 (HR, α-9)	−	16	Negative	Negative
24	42	51 (HR, α-5)	−	51	Negative	Negative
25	30	−	40 (LR, α-8)	39	39, 40	39, 40
26	45	31 (HR, α-9)	−	31,74	74	74
27	28	16 (HR, α-9)	−	16, 52, 56, 55, 74, 81	52, 56, 55, 74, 81	52, 56, 55, 74, 81
28	43	58 (HR, α-9)	−	52, 58	52	52
29	56	16 (HR, α-9)	−	16, 42	42	42
30	53	−	58 (HR, α-9), 82 (pHR, α-5)	45, 51, 90	45, 51, 58, 82, 90	90
31	25	56 (HR, α-6)	−	56, 34	34	34
32	32	16 (HR, α-9)	−	16	Negative	Negative
33	46	52 (HR, α-9)	−	52	Negative	Negative
34	53	51 (HR, α-5), 82 (pHR, α-5)	−	51, 82	Negative	Negative
35	34	82 (pHR, α-5)	−	51, 82	51	51
36	53	52 (HR, α-9)	−	52	Negative	Negative
37	29	52 (HR, α-9)	−	33, 52	33	Negative
38	48	−	62 (LR, α-3)	52, 53, 81	52, 53, 62, 81	53, 62
39	51	58 (HR, α-9)	−	51, 58	51	51
40	32	−	81 (LR, α-3)	31	31, 81	31, 81
41	31	−	42 (LR, α-1)	58	58	58, 42
42	50	66 (pHR, α-6)	−	40, 66	40	Negative
43	51	51 (HR, α-5)	−	51, 56, 58	56, 58	56
44	40	56 (HR, α-6)	−	56	Negative	Negative
45	49	67 (pHR, α-9)	−	67	Negative	Negative
46	48	52 (HR, α-9)	−	52	Negative	Negative
47	59	−	74 (LR, α-10)	44	44, 74	44
48	51	−	11 (LR, α-10)	35, 67	35, 11, 67	35
49	33	31 (HR, α-9)	−	31, 58, 74	58, 74	58, 74
50	26	−	55 (LR, α-10)	52, 56, 59, 30, 54, 74	52, 56, 59, 30, 54, 55, 74	52, 56, 59, 30, 54, 55, 74
51	37	52 (HR, α-9)	−	52	Negative	Negative
52	37	−	42 (LR, α-1), 62 (LR, α-3)	51, 53, 82	51, 42, 53, 62, 82	51, 42, 53, 62, 82
53	54	31 (HR, α-9)	58 (HR, α-9), 66 (pHR, α-6)	31, 51, 82, 6b	51, 58, 6b, 82	6b, 66, 82
54	57	16 (HR, α-9)	−	16, 42	42	42
55	31	52 (HR, α-9)	−	31, 52, 58, 70	31, 58, 70	31, 58, 70
56	40	52 (HR, α-9)	−	52	Negative	Negative
57	51	82 (pHR, α-5)	−	51, 58, 82	51, 58	51, 58
58	65	53 (pHR, α-6)	−	53	Negative	Negative
59	28	−	53 (pHR, α-6), 74 (LR, α-10)	16, 52	16, 52, 53, 74	16, 52, 53, 74
60	46	−	74 (LR, α-10)	31	31, 74	74
61	30	66 (pHR, α-6)	−	53, 66	53	53
62	37	52 (HR, α-9)	−	52	Negative	Negative
63	32	31 (HR, α-9)	−	31	Negative	Negative
64	25	−	84 (LR, α-3)	Negative	84	84
65	38	18 (HR, α-7)	−	18	Negative	Negative
66	47	−	66 (pHR, α-6), 90 (LR, α-15)	16	16, 66, 90	Negative
67	54	52 (HR, α-9)	−	52	Negative	Negative
68	28	−	90 (LR, α-15)	42	42	42, 90
69	40	16 (HR, α-9)	59 (HR, α-7)	16, 74	59, 74	74
70	43	−	53 (pHR, α-6)	18	18, 53	18, 53
71	33	51 (HR, α-5)	−	51, 74	74	74
72	42	−	67 (pHR, α-9)	Negative	Negative	67
73	30	−	74 (LR, α-10)	16, 39	16, 39	16, 39, 74
74	49	67 (pHR, α-9)	−	67	Negative	Negative
75	26	56 (HR, α-6)	−	56	Negative	Negative
76	34	51 (HR, α-5)	−	51	Negative	Negative
77	27	−	54 (LR, α-13)	34	34, 54	34, 54
78	50	66 (pHR, α-6)	−	40, 66	40	40
79	46	−	73 (pHR, α-11)	54, 61	61, 73	54,61, 73
80	51	58 (HR, α-9)	−	51, 56, 58	51, 56	51, 56
81	35	26 (pHR, α-5)	−	26, 90	90	Negative
82	56	−	89 (LR, α-3)	Negative	Negative	89
83	26	52 (HR, α-9)	−	52, 59	59	Negative
84	29	33(HR, α-9), 52 (HR, α-9)	−	33, 52	Negative	Negative
85	28	−	42 (LR, α-1)	16, 54, 82	16, 42, 54, 82	16, 54, 82
86	27	18 (HR, α-7), 59 (HR, α-7)	−	18, 31, 52, 59, 55, 67, 74, 90	31, 52, 55, 67, 74, 90	31, 55, 67, 74, 90

In the three columns on the right, all HPV types detected in cervical and vaginal samples are compared, and HPV types detected in only one of the samples are defined as the HPV type that preferentially infected that site. The HPV types per case are shown in the order of HR type and then other type. In the left column, HPV types are indicated for the preferentially infected sites. HPV, human papillomavirus; LR, low-risk; HR, high-risk; pHR, possible high-risk.

**Table 3 ijms-24-13151-t003:** Colocalization rate (including possible contamination) of different HPV risk groups in the cervical samples or in the vaginal samples in 178 patients.

Combination of Infecting HPV Types			Samples (n)	Colocalization Rate by LR Types in Cervical Samples	Colocalization Rate by HR Types in Vaginal Samples
LR			18	34.8%	−
LR		+pHR	8		
LR	+HR		24		73.6%
LR	+HR	+pHR	12		
	HR		79	−	
	HR	+pHR	16		
		pHR	21		−

HPV, human papillomavirus; LR, low-risk; HR, high-risk; pHR, possible high-risk.

## Data Availability

The data and material that support the findings of this study are available from the corresponding author upon reasonable request.

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
