# Peer review of "Preferential Tissue Sites of Different Cancer-Risk Groups of Human Papillomaviruses"

_ijms, 2023, doi:10.3390/ijms241713151_

Round 1

Reviewer 1 Report

This is an interesting work that has an adequate design to avoid contamination when taking samples from the areas of interest.

Samples were carefully obtained independently from the cervix, upper vagina or lower vagina from patients referred to colposcopy. The HPV typing was performed in 264 patients that were HPV-positive in any of the three samples, with a PCR method which detects 39 mucosal HPV types (12 HR, 12 pHR and 15 LR) by the amplification of E6 and E7 genes with specific primers. Results were analyzed according to the individual HPV type, HPV risk group, and the alpha phylogenetic groups. It is interesting that HPV types most frequently found in cervix are HPV52, 51, and in third place HPV16 and 58.

The preference of the HPV types for cervix or for vagina was evaluated in 86 patients who presented inconsistent HPVs in cervix vs vagina. I consider it ingenious that in this analysis they considered a viral type with preference in one site when it was only found in that site, since if, for example, it was found in both the cervix and the vagina, a carryover or contamination of one of the sites could have occurred.

HR HPV types were most commonly found in cervix (87% of the cases) than in vagina (6.7-fold most frequent in cervix), while LR types were found in vagina. There were no differences in site preference of the pHRs types.

I consider that this work provides valuable information in the field, it is well written, has a fine analysis, the information in the tables is clearly presented, the discussion is well performed and limitations are clearly stated.

My only comment is in relation to the analysis of table 4 (colocalization rate), that even it is well discussed at the discussion section, the intention of the analysis and meaning of the findings could be more clearly explained at the results.

Reviewer 2 Report

The article is devoted to an extremely interesting problem - clarifying the biological basis of the oncogenic potential of different genotypes of human papillomavirus. The aspect associated with the tropism of different genotypes to different cell types of the cervical and vaginal epithelium is not well understood. The reason for this is the technical complexity of organizing the collection of material from different localizations, excluding cross-contamination with cellular material from these localizations. Judging by the results presented, the authors succeeded. The work was performed methodologically carefully, the result obtained is valuable. The sample analyzed is sufficient to formulate statistically valid conclusions, which are formulated carefully, with all the necessary reservations. A literature review provides an overview of the current state of knowledge on the issues under discussion. The manuscript is written in clear language. I believe that the article should be published.

To the relative disadvantages of the manuscript, I would refer to the presentation of results exclusively in the form of cumbersome tables. Presenting at least part of the results graphically, for example, in the form of diagrams, in my opinion, could facilitate the reader's perception of the article. Information on the relative viral loads of different genotypes of papillomavirus in different localizations could also be useful (the genotyping method used by the authors, unfortunately, does not provide such information). In most cases, the same genotype of the virus was still found in the vagina and in the cervix, in which case it would be interesting to compare the loads.

The statement "The genotypes of pHR HPVs are undetected in normal tissues" (line 169) is questionable, I would suggest softening the wording.
